# Fabrication of Periodic Nanostructures on Silicon Suboxide Films with Plasmonic Near-Field Ablation Induced by Low-Fluence Femtosecond Laser Pulses

**DOI:** 10.3390/nano10081495

**Published:** 2020-07-30

**Authors:** Tatsuyoshi Takaya, Godai Miyaji, Issei Takahashi, Lukas Janos Richter, Jürgen Ihlemann

**Affiliations:** 1Department of Applied Physics, Tokyo University of Agriculture and Technology, 2-24-16 Nakacho, Koganei, Tokyo 184-8588, Japan; s190953t@st.go.tuat.ac.jp (T.T.); s202053s@st.go.tuat.ac.jp (I.T.); 2Laser-Laboratorium Göttingen e. V., Hans-Adolf-Krebs-Weg 1, 37077 Göttingen, Germany; lukas.richter@llg-ev.de

**Keywords:** femtosecond laser, laser ablation, nanostructure formation, surface plasmon polaritons, near-field, silicon suboxide, glass

## Abstract

Silicon suboxide (SiO*_x_*, *x* ≈ 1) is a substoichiometric silicon oxide with a large refractive index and optical absorption coefficient that oxidizes to silica (SiO_2_) by annealing in air at ~1000 °C. We demonstrate that nanostructures with a groove period of 200–330 nm can be formed in air on a silicon suboxide film with 800 nm, 100 fs, and 10 Hz laser pulses at a fluence an order of magnitude lower than that needed for glass materials such as fused silica and borosilicate glass. Experimental results show that high-density electrons can be produced with low-fluence femtosecond laser pulses, and plasmonic near-fields are subsequently excited to create nanostructures on the surface because silicon suboxide has a larger optical absorption coefficient than glass. Calculations using a model target reproduce the observed groove periods well and explain the mechanism of the nanostructure formation.

## 1. Introduction

Structures smaller than the wavelength of light can induce optical anisotropy and rotatory and resonant scattering [1,2]. Recent developments in material nanoprocessing techniques enabled the regular arrangement of nanostructures on or inside solids to perform many attractive applications such as optical cloaking [3,4], photon trapping [5,6], and structured light generation [7,8]. Because glass is transparent in the visible region, chemically stable, and inexpensive, it is a promising material for use in these applications, and it is popularly used for many kinds of optical elements and optical integrated devices.

Periodic nanostructures with a sub-μm groove period *d* can be easily formed inside or on glass by superimposed multiple shots of tightly focused femtosecond (fs) laser pulses of a few μm in diameter and at a fluence of a few to 10 J/cm^2^ by using a high-numerical-aperture (NA) lens such as a microscope objective [9,10,11]. Recently, using the birefringence of nanostructures formed in fused silica with fs pulses, Beresna et al. developed a spatial-distributed wave plate that can convert a Gaussian beam to structured light such as a radially polarized beam or an optical vortex [12]. This remarkable optical element is currently commercially available. Since laser nanoprocessing can be rapidly applied to a large area, it has attracted much interest as an industrial nanofabrication technique [13]. However, the focal spots of a few μm and a short focusing depth being required for nanostructuring somewhat restrict the possible applications. Current limitations include slow processing times and short working distances between the target and the focusing lens [14].

Silicon suboxide is a substoichiometric silicon oxide that has a larger refractive index and a larger optical absorption coefficient in the region from ultra-violet (UV) to near-infrared (NIR) than other glass materials such as fused silica, borosilicate, and soda-lime glass [15,16,17,18]. In addition, it can be easily oxidized to SiO_2_ by thermal treatment (~1000 °C) in air to become transparent in the UV–NIR region [18]. Thermal treatment in an oxygen-free environment leads to the formation of silicon nanocrystals [19]. As a result of these properties, this material has attracted attention for applications such as anti-reflection coating [20], photo luminescence [21], giant Raman scattering [22], and SiO_2_ precursors for laser processing [23].

In this paper, we describe the successful formation of a nanostructure with a groove period of *d* = 200–330 nm in air on a silicon suboxide (SiO*_x_*, *x* ≈ 1) surface irradiated with 800 nm, 100 fs laser pulses at ~700 mJ/cm^2^ using a low NA lens. Preliminary results were presented in [24]. The spot size was 120 μm in diameter and the fluence was an order of magnitude lower than that needed for structuring glass materials. However, thus far, the fs-laser-induced nanostructuring of silicon suboxide has not been well understood, and its successful control has not been achieved. In this work, based on the formation process of a nanostructure observed on silicon suboxide, we discuss its physical mechanism. The groove period calculated for a model target closely matches the observed groove period. These results showed that the formation of a thin layer of high-density electrons and the excitation of surface plasmon polaritons (SPPs) are responsible for the nanostructure formation on a silicon suboxide induced with intense femtosecond laser pulses.

## 2. Experimental

As a target, we used a silicon suboxide, SiO*_x_* (*x* ≈ 1) film of 1.4 μm thickness deposited on a fused silica substrate by thermal evaporation (Leybold UNIVEX 350, Cologne, Germany) [18]. Figure 1a shows the refractive index *n* and the extinction coefficient *κ* measured as a function of wavelength *λ* = 300–900 nm with a spectroscopic ellipsometer (M-220, JASCO Corporation, Tokyo, Japan). As seen in Figure 1b, the film is colored yellow because *κ* is especially large in the UV to blue region. The nonlinear absorption coefficient *β* of the film was not measured in this work, though values of *β* = 10^−7^–10^−5^ cm/W were measured using an 800 nm fs laser, which are much larger than the value for fused silica [16,17]. In addition, because the refractive index of silicon suboxide was measured as *n* = 1.8–2.4 in the UV–NIR region, which is much larger than that of glass such as fused silica as shown in Figure 1a, *β* of the film applied in this work should also be larger. In a previous report, a silicon suboxide film 700 nm thick was confirmed to be able to oxidize to SiO_2_ by annealing in air at ~1000 °C for 48 h, resulting in an increase of the transmittance in the UV–NIR region [18].

Figure 2a shows a schematic diagram of the optical configuration used in the ablation experiment. We used 800 nm and 100 fs laser pulses delivered from a Ti:sapphire laser system at a repetition rate of 10 Hz. The incident laser pulses were controlled with a mechanical shutter to be either a superimposed number of pulses of *N* = 250–1000 or a single shot *N* = 1. The pulse energy was controlled with a pair of a half-wave plate and a polarizer to set the laser fluence to *F* = 625–750 mJ/cm^2^. To measure the energy shot by shot, we acquired the energy of the pulse reflected at a glass surface with a photodiode. The linearly polarized fs laser pulses were focused in air at normal incidence on the silicon suboxide films with a lens with a 250 mm focal length. To monitor the surface irradiated by the pulses, the microscopic image of the target surface was observed with a charge-coupled device (CCD) camera with a He–Ne laser beam used for illumination. For measuring the intensity profile of the focal spot, we tilted the target to observe the fs pulse reflected at the surface with the camera. The result is shown in Figure 2b. The intensity distribution of the focal spot fitted the lowest-order Gaussian profile well. The focal spot size was 120 μm in diameter at 1/*e*^2^ of the maximum intensity.

The surface morphology of the target was observed with an optical microscope (VH-Z500R, Keyence, Osaka, Japan) and a scanning electron microscope (SEM, JSM-6510, JEOL, Tokyo, Japan). A 10 nm thick platinum layer was applied homogeneously onto the irradiated surface with an ion sputter coater to improve the conductivity of the surface, thus improving the clarity of the SEM image. To evaluate the spatial periodicity of the surface structure, a two-dimensional Fourier transform (using the image processing software, SPIP, Image Metrology, Lyngby, Denmark) was performed on the SEM image in a 5 × 5 μm^2^ region to acquire the spatial frequency distribution along the polarization direction.

## 3. Results and Discussion

The nanostructures formed on various kinds of materials irradiated with fs laser pulses at a fluence *F* lower than the single-shot ablation threshold *F*_1_ [25]. To compare the *F*_1_ of the silicon suboxide film to those of the glass materials [26,27], we measured *F*_1_ from the optical microscope images of the single-shot ablation traces. Figure 3a shows an example of an optical microscope image of the film surface irradiated with a single fs pulse at *F* = 1200 mJ/cm^2^. The light gray area in the center of the image is the ablated trace. Then, while varying *F*, we measured the area S of the corresponding ablation traces. Figure 3b shows *S* plotted as a function of *F*. By extrapolating the fitting line (solid line in Figure 3b), the single-shot ablation threshold of the silicon suboxide was estimated to be *F*_1_ = 1060 (±10) mJ/cm^2^, which is much smaller than *F*_1_ = 2 J/cm^2^ for fused silica [26], *F*_1_ = 4.1 J/cm^2^ for borosilicate glass [27], and *F*_1_ = 3.4 J/cm^2^ for soda-lime-silicate glass [27]. The results showed that the small *F*_1_ for silicon suboxide could be attributed to the large optical absorption coefficient [15].

Figure 4 shows an SEM image and the spatial frequency spectrum of the silicon suboxide film irradiated at *F* = 675 mJ/cm^2^ for *N* = 250, 500, and 1000 pulses. For *N* ≤ 200, no ablation trace was observed. Increasing *N* to 250, ablation traces were observed on the surface at several places in the focal spot. As shown in Figure 4a, dot-like nanostructures formed over the whole area of the traces, while line-like nanostructures expanding perpendicular to the polarization direction were generated at the center of the traces. With a further increase of *N* to 500 and 1000, as seen in Figure 4b,c, the area over which the dot-like nanostructures formed extended more widely than that for the line-like nanostructures. In the spatial-frequency spectrum, therefore, the spectral peak indicating the periodic structures in the ablation trace could not be identified.

An increase in *F* was expected to increase the density of the free electrons produced in the surface layer to a level sufficient to change the surface morphology. To confirm this, we irradiated the film surfaces with fs pulses at *F* = 700 mJ/cm^2^. The results are shown in Figure 5. At *N* = 250 (Figure 5a), the line-like nanostructures clearly formed with a period of ~160 nm in a larger area than those generated at *F* = 675 mJ/cm^2^. Increasing *N* to 500 (Figure 5b), the area over which both the dot-like and line-like nanostructures formed widened from the center of the focal spot to the edge. At *N* = 1000, as shown in Figure 5c, the spectral peak of the ablation trace appeared faintly at *d* ≈ 220 nm, corresponding to the peak at 4.5 μm^−1^.

We increased *F* to 750 mJ/cm^2^, and observed the surface morphology of the films irradiated with the fs pulses. The results are shown in Figure 6. At *N* = 250 (Figure 6a), the line-like nanostructures clearly formed with a spectral peak at *d* = 210–320 nm, where *d* is defined as the full width at a half maximum of the spectrum with the background signal subtracted (dashed line in Figure 6). With increasing *N*, the line-like nanostructures were formed with *d* = 230–290 nm at *N* = 500 (Figure 6b) and *d* = 200–330 nm at *N* = 1000 (Figure 6c), where the other spectral peaks at the respective harmonic frequencies were virtual. In this experiment, we could not clearly observe any change in the spectral peak position for increasing *N*. At *N* = 1000, the nanostructures homogeneously formed in an area of ~20 μm in diameter in the center of the focal spot, which is an order of magnitude larger than that on glass irradiated with tightly focused fs pulses. These results showed that fs laser pulses are strongly absorbed in a silicon suboxide surface to produce high-density electrons in the vicinity of the surface, leading to nanoablation with intense near-fields of SPPs.

Based on the experimental results and the calculation using a model target, we discussed the excitation of SPPs in the silicon suboxide surface with the fs pulses and the subsequent nanostructure formation. The reflectivity of 1.4% at the interface between SiO*_x_* and substrate was calculated at normal incidence. Assuming the constructive interference between the incident and reflected pulses [28], the fluence at the surface could be enhanced by a factor of 1.25, leading to a decrease in the effective *F*_1_ from 1060 to 850 mJ/cm^2^. The experimental results for *F* = 675–750 mJ/cm^2^, as shown in Figure 4, Figure 5 and Figure 6, clearly showed that these fluences are lower than the single-shot ablation threshold and higher than the multiple-shot ablation threshold for the pulses with *N* ≥ 250. Assuming that the superimposed multiple fs laser pulses at a fluence lower than the single-shot ablation threshold induce surface modifications within a thin surface layer [29,30,31], subsequent fs pulses could induce a high density of electrons suitable to excite SPPs in this layer [32,33]. As shown in a previous report on pump-probe reflectivity measurements, an intense fs laser pulse can produce electrons with a high density of ~10^22^ cm^−3^ in fused silica [34]. Assuming the production of the high-density electrons in the silicon suboxide, the permittivity of the silicon suboxide can be described using a Drude model [35,36]:(1)$$\varepsilon_{a}=\varepsilon_{\mathrm{siox}}-\omega_{p}^{2}/\left(\omega^{2}+i\omega /\tau \right),$$
where *ε*_siox_ = 3.24 is the permittivity of the silicon suboxide at *λ* = 800 nm measured with the ellipsometer, and the second term represents the modulation by free-carrier response at the electron density *N*_e_ produced in the silicon suboxide surface. Here, *ω* = 2π*c*/*λ* is the laser frequency in vacuum, *c* is the light speed in vacuum, *τ* is the Drude damping time, and *ω*_p_ = (*e*^2^*N*_e_/(*ε*_0_*m***m*))^/2^ is the plasma frequency with the permittivity of vacuum *ε*_0_, the electron charge *e* and mass *m*, and the optical effective mass of an electron *m**. We ignored other effects modulating the permittivity such as band and state filling [37,38] and band gap renormalization [39,40,41] because they are very small.

The calculation method to determine the plasmon wavelength *λ*_spp_ in the surface layer irradiated with the fs pulse is almost the same as that used in previous studies [32,33]. Briefly, *λ*_spp_ = 2π/Re[*k*_spp_] was calculated using the following relation between light and SPPs:(2)$$k_{\mathrm{spp}}=k_{0}\sqrt{\frac{\varepsilon_{a}\;\varepsilon_{b}}{\varepsilon_{a}+\varepsilon_{b}}},$$
where *k*_0_ = 2π/*λ* is the laser wavenumber in vacuum. Assuming that the SPPs are excited at the interface between the silicon suboxide and the surface layer with high-density electrons produced with the fs laser pulse, as shown in the inset of Figure 7, we set *ε*_b_ = *ε*_siox_. For the excitation of SPPs, that is, for evanescent waves to exist in the vicinity of the surface, the relation Re[*ε*_a_] < 0 should be satisfied [42]. The excitation of the SPPs is the origin of the periodicity of the fs laser-induced nanostructure formation, and periodic nanoablation is induced by a fine spatial distribution of electromagnetic energy in the surface layer [32,33,43,44,45,46,47,48,49,50,51]. To form the stationary energy distribution, the following two processes were proposed: the interference between the incident laser beam and the SPPs [43,44,45], and the counter-propagating SPP interference, i.e., the generation of a standing wave mode of SPPs [32,33,46,47,48,49,50,51]. The groove periods of the aforementioned types of interference are *λ*_SPP_ and *λ*_SPP_/2, respectively. Assuming that either these types of interference occur simultaneously or the latter occurs dominantly, the period could become *λ*_SPP_/2.

Figure 7 shows the calculated groove period *D* plotted as a function of *N*_e_. We reported the damping time and optical effective mass of an electron to be *τ* = 0.1–2 fs [52,53] and *m** = 1, respectively. Here, we show two results at *τ* = 0.5 fs (thick red curve) and *τ* = 1 fs (thin blue curve) for *m** = 1. Excitation of SPPs is allowed in the region Re[*ε*_a_] < 0, as shown by the solid curves in Figure 7. For both values of *τ*, when the high-density electrons are excited in the surface layer and metalize the surface with the fs pulse, the calculated period is *D* = 180–430 nm. At *τ* < 0.5 fs, *D* was calculated to approach ~200 nm. If the SPPs are resonantly excited, where *ε*_a_ + *ε*_b_ becomes zero [42,50], *D* could be 267 nm for *τ* = 1 fs at *N*_e_ = 1.0 × 10^22^ cm^−3^. Here, *N*_e_ for these values of *D* is much larger than the critical plasma density of 1.7 × 10^21^ cm^−3^. *N*_e_ can reach a higher value than this critical density because the electrons are generated in the solid surface [34,54]. These calculated results are in good agreement with the observed period *d*. The results showed that plasmonic near-fields generated in the silicon suboxide surface induce nanoablation to form a periodic nanostructure.

## 4. Conclusions

We found that clear periodic nanostructures can be formed on a silicon suboxide film with superimposed fs laser pulses at low fluence. The experimental and calculated results obtained showed that the low-fluence fs pulses are absorbed near the surface due to the large absorption coefficient of the film, forming a thin layer having high-density electrons and leading to nanoablation by plasmonic near-fields. The interfering fs pulses can form grooves with a period of λ/(2sinθ), where *θ* is the incident angle of the laser pulse [55]. By using a high-NA optical configuration, the period can reach λ/2. For nanostructuring in this present work, the period *λ*_SPP_/2 was much smaller than *λ*/2, as denoted in Equation (2). As silicon suboxide can easily be transformed into transparent glass, the proposed technique should provide a useful approach for rapidly and homogeneously fabricating nanostructures on glass.

## Figures and Tables

**Figure 1 nanomaterials-10-01495-f001:**
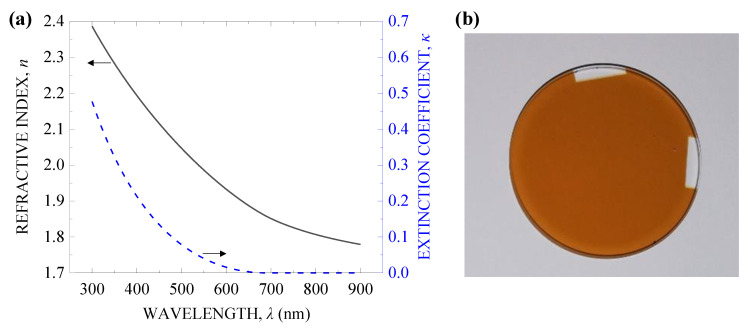
(**a**) Refractive index *n* (black solid line) and extinction coefficient *κ* (blue dashed line) of the silicon suboxide film measured as a function of the wavelength *λ*. (**b**) Photograph of the film on a fused silica substrate 25 mm in diameter.

**Figure 2 nanomaterials-10-01495-f002:**
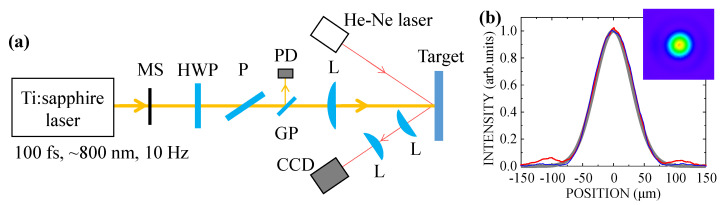
(**a**) Schematic diagram of the optical configuration for the ablation experiment. MS, mechanical shutter; HWP, half-wave plate; P, polarizer; GP, glass plate; PD, photodetector; L, lens. (**b**) Intensity profile of the focal spot of a fs pulse. The red and blue curves represent the horizontal and vertical profiles, respectively, with a Gaussian profile shown in gray for comparison. The inset shows a CCD image of the focal spot.

**Figure 3 nanomaterials-10-01495-f003:**
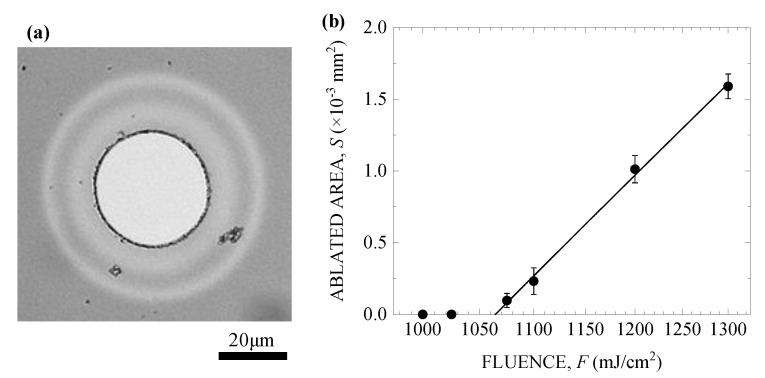
(**a**) Optical microscope image of the silicon suboxide film irradiated with a single fs pulse at *F* = 1200 mJ/cm^2^, and (**b**) single-shot fs laser ablated area *S* plotted as a function of the laser fluence *F*.

**Figure 4 nanomaterials-10-01495-f004:**
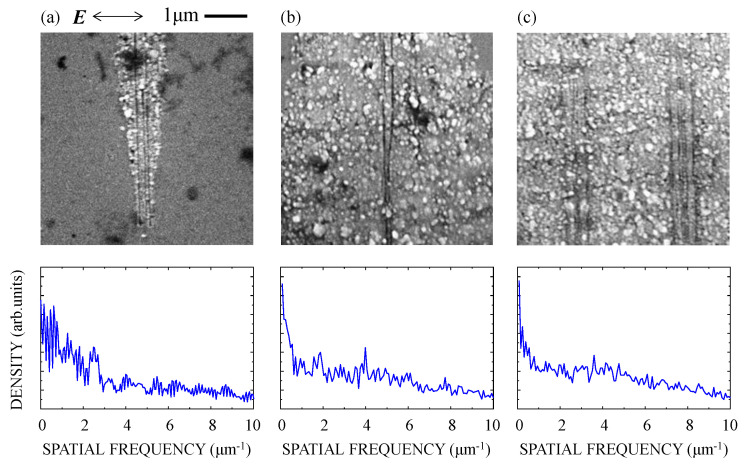
SEM image (top) and Fourier spectrum (bottom) of the silicon suboxide films irradiated with (**a**) *N* = 250, (**b**) *N* = 500, and (**c**) *N* = 1000 fs laser pulses at *F* = 675 mJ/cm^2^. ***E*** denotes the direction of the polarization.

**Figure 5 nanomaterials-10-01495-f005:**
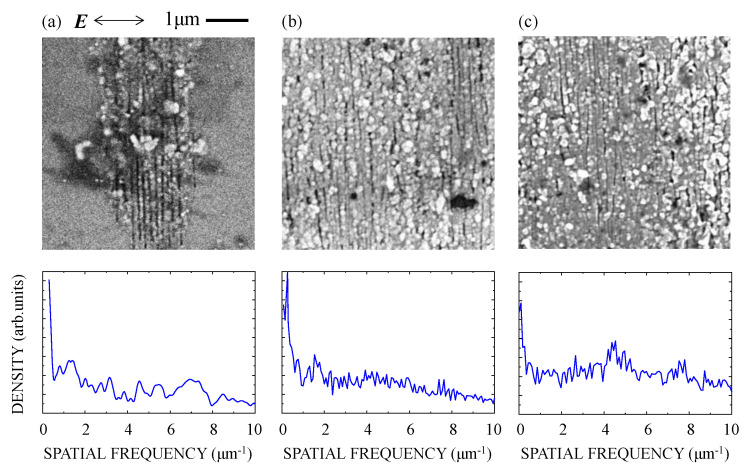
SEM image (top) and Fourier spectrum (bottom) of the silicon suboxide films irradiated with (**a**) *N* = 250, (**b**) *N* = 500, and (**c**) *N* = 1000 fs laser pulses at *F* = 700 mJ/cm^2^. ***E*** denotes the direction of the polarization.

**Figure 6 nanomaterials-10-01495-f006:**
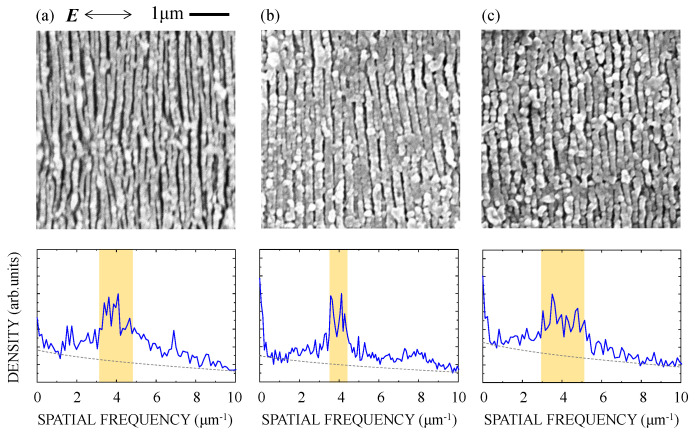
SEM image (upper) and Fourier spectrum (lower) of the silicon suboxide films irradiated with (**a**) *N* = 250, (**b**) *N* = 500, and (**c**) *N* = 1000 fs laser pulses at *F* = 750 mJ/cm^2^. ***E*** denotes the direction of the polarization. The period *d* is estimated from the half maximum of the spectrum (orange-hatched area in the spectrum).

**Figure 7 nanomaterials-10-01495-f007:**
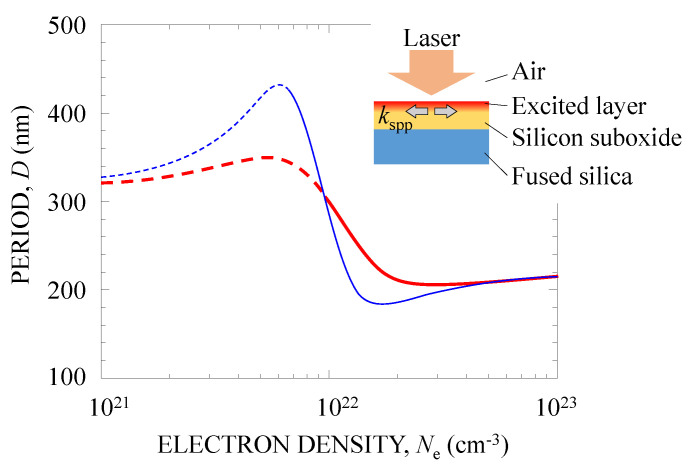
Period *D* of nanostructures calculated as a function of electron density *N*_e_. The thin blue and thick red curves denote *D* at the damping times of *τ* = 1 fs and *τ* = 0.5 fs, respectively. Excitation of SPPs is allowed in the region Re[*ε*_a_] < 0, where *D* is drawn with solid lines.

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
