# Peer review of "Fabrication of Periodic Nanostructures on Silicon Suboxide Films with Plasmonic Near-Field Ablation Induced by Low-Fluence Femtosecond Laser Pulses"

_nanomaterials, 2020, doi:10.3390/nano10081495_

Round 1

Reviewer 1 Report

Authors: Tatsutoshi Takaya, Godai Miyaji, Issei Takahashi, Lukas Janos Richter, Jürgen Ihlemann

Title: Fabrication of periodic nanostructures on silicon suboxide films with plasmonic near-field ablation induced by low-fluence femtosecond laser pulses

SUMMARY

The paper present a typical experiment of LIPSS formation on a dielectric sample of SiOx deposited on a substrate of Si. The obtained periodicity is close to the one that was already observed in a number of dielectrics, positioning this article as an incremental contribution. Some of the authors in the past have published the same kind of results and seem to be repeating similar research using another material: the SiOx. Therefore, the novelty of the proposed study is low. Moreover, the theoretical analysis is inherited from past mistakes that were not corrected by authors of similar studies. To avoid spreading errors, detailed explanation aiming at helping the authors is provided below. If editor would really like to publish this work, it is encouraged to remove the theoretical explanations provided by the authors.

SUGGESTION: Major revision or reject

On the artificial division by a factor 2 of the period computed from Eq. (2).

The response of the authors to this point was unsatisfactory. The references cited by the authors in their response letter are mostly self-citations that do not address the problem adequately, and other papers did not address the reason how an artificial division of period by 2 could be justified. Particularly, Ref. 47 invokes the Bragg scattering law that applies for isotropic scattering of beams on atoms (https://en.wikipedia.org/wiki/Bragg%27s_law), but not on light anisotropic scattering on roughness / nanoparticles, as it is the case in the LIPSS formation plasmonic mechanism. Authors may understand that coherent light scattering does not take place in isotropic manner [Novotny, L.; Hecht, B. & Pohl, D. W. Interference of locally excited surface plasmons. Journal of Applied Physics, 1997, 81, 1798].

Moreover, authors claimed that two types of SPP exist: those of period scaled with wavelength, and those of period scaling with lambda/2. This response rather characterizes a misunderstanding of the SPP phenomenon, since (i) the period of the SPP strongly depends on the topography of the surface; (ii) the two main surface modes induced by coupling of laser light with roughness are scaling with the wavelength, and with wavelength divided by optical refractive index [45].

To help authors realizing the problem further: when writing about the “counter-propagating wave” that attempt to justify the factor 2, which direction of propagation is considered in respect of the surface topography? Another way to express it is: how many scattering centers are employed by authors in their reasoning for coupling the far field laser wave with the surface electromagnetic SPP modes? Zero? One? Or many?

Let’s study these possible answers:
- “
Zero”: this case corresponds to the ideal model that authors employed in the paper (Eq. 2). In that case, the invoked counter-propagative wave was introduced in a Helmholtz equation (see Ref. [44]) prior to deducing the dispersion relation indicated by authors in Eq. (2). Therefore, any concept of forward and counter-propagating wave interference are already included in the Eq. (2) itself, without having to make an artificial post-derivation division by 2.

- “One”: this case corresponds to the problem of a laser wave scattering on an isolated nano-object, e.g., a nanoparticle, or a ridge. In that case, “counter-propagative wave” does not meet any forward-propagating wave, then a division of the period by 2 is also not justified.

- “Many” branches to the case of scattering of the laser wave on a collection of scattering centers that interact together, e.g., the surface roughness, or a collection of particles, etc; that is already well included in the model of Sipe et al, where the collection of scattering centers do interact explicitly. Although being classical, this paper was not even not cited by the authors [Sipe, J. E.; Young, J. F.; Preston, J. & van Driel, H. M. Laser-induced periodic surface structure. I. Theory. Phys. Rev. B, 1983, 27, 1141-1154.]. Moreover, the later model was proven to be well grounded by numerous authors, e.g.,
- Dufft, D.
et al., Femtosecond laser-induced periodic surface structures revisited - A comparative study on ZnO, J. Appl. Phys., 2009, 105, 034908.
- Skolski, J. et al., Laser-induced periodic surface structures: Fingerprints of light localization
Physical Review B, 2012, 85, 075320;
- Hao Zhang, Jean-Philippe Colombier, Stefan Witte, Laser-induced periodic surface structures: Arbitrary angles of incidence and polarization states,
Physical Review B, 2020, 101, 245430.

In each of the presented sub-cases, the “counter-propagative wave” invoked by authors has a different meaning and different consequences on the periodicity. In addition, the interference of the surface modes was already accounted in each of these descriptions.

As a conclusion of this comment, it is important to understand that artificially dividing the period obtained from a simple formula by a factor 2 (Bragg condition is hereby not applicable) is not grounded.

If editor would really like to publish this work, it is proposed to remove the theoretical explanations provided by the authors.

Reviewer 2 Report

Authors addressed most of Reviewer's comments and the manuscript is nou suitable for publication.

Reviewer 3 Report

Revision have answered the raised issues.

Reviewer 4 Report

Authors in their revised manuscript of collaborative research 867156 present novel results on the periodic structure fabrication via femtosecond laser processing. They have performed a tremendous job in the revising the manuscript according to most of the comments declared by 4 reviewers and did the perfect job by answering point by point questions. Now the paper is very-well-constructed thus deserves publication. My recommendation is ACCEPT for publication in the Nanomaterials journal by MDPI.

Minor cosmetic corrections to reference list:

  1. I recommend remove reference [25] from the text of the manuscript, and incorporate the text “We used the image processing software (SPIP, Image Metrology).“ to the main text of the manuscript.
  2. The same applies to reference [29]. I recommend remove it from text and incorporate the text “The reflectivity of 1.4% at the interface between SiOx and substrate was calculated at normal incidence. Assuming the constructive interference between the incident and reflected pulses [30], the fluence at the surface could be enhanced by a factor of 1.25. Even if the effective F1 decreases from 1060 mJ/cm2 to 850 mJ/cm2, the fluence is still higher than that needed for nanostructure formation.” directly to the manuscript.
  3. I recommend removing the text „See, for example,” from references [33], [34].
  4. I recommend removing reference [49] form the text of the manuscript and incorporate the text “The interfering fs pulses can form grooves with a period of λ/(2sinθ), where θ is the incident angle of the laser pulse, as reported in Ref. 50. By using a high-NA optical configuration, the period can be reached to λ/2. For nanostructuring in this present work, the period λSPP/2 is much smaller than λ/2, as denoted in Eq. (2) “ directly to the text of the manuscript.
